# Three-dimensional discrete element simulations on pressure ridge formation

Marek Muchow[1] and Arttu Polojärvi[1]

[1]Aalto University, School of Engineering, Department of Mechanical Engineering, P.O. Box 14100, FI-00076 Aalto, Finland

**Correspondence:** Marek Muchow (marek.muchow@aalto.fi)

**Abstract.** This study presents the first three-dimensional discrete element method simulations of pressure ridge formation. Pressure ridges are an important feature of the sea-ice cover, as they contribute to the mechanical thickening of ice and likely limit the strength of sea ice at large scale. We validate the simulations against laboratory-scale experiments, confirming their accuracy in predicting ridging forces and ridge geometries. Then we demonstrate that Cauchy-Froude scaling applies for translating laboratory-scale results on ridging to full-scale scenarios. We show that non-simultaneous failure, where an ice floe fails at distinct locations across the ridge length, is required for an accurate representation of the ridging process. This process cannot be described by two-dimensional simulations. We also find a linear relationship between the ridging forces and the ice thickness, contrasting with earlier results in the literature obtained by two-dimensional simulations.

## 1 Introduction

In this study, we simulate pressure ridge formation by using a three-dimensional discrete element method (DEM) (Cundall and Strack, 1979) model for the first time. Pressure ridging is an ice failure process resulting from relative compression of two or more ice floes driven by winds and currents. Ridges may also form as a result of failure of an intact ice sheet. Ridges consist of ice rubble formed by ice fragments accumulated in a keel underwater, and a sail on top of the ice (Figure 1) and may partly consolidate over time. During the ridge formation process, however, continuous ice rubble deformation is likely to inhibit consolidation. Ridging is assumed to be one of the main mechanisms limiting large-scale strength of sea ice (Lipscomb et al., 2007), it influences the local ice thickness (Lepäranta and Hakala, 1992), and it increases the overall sea-ice volume (Itkin et al., 2018; von Albedyll et al., 2022). Therefore, ridging has a major role in sea-ice redistribution in Earth System Models (Thorndike et al., 1975; Lipscomb et al., 2007). Understanding ridging processes is, thus, of utmost importance for sea-ice dynamics. Additionally, understanding ridging processes is important to resolve ridges accurately in high-resolution forecasting simulations as needed, for example, for planning ship routes and locations for offshore wind farms.

The first theoretical models for ridging were established in the 1970s with the kinematic ridging model by Parmerter and Coon (1972). More recent developments focus on the representation of ridging in Earth System models (Roberts et al., 2019).

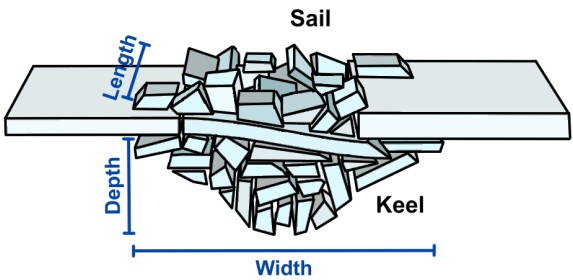

**Figure 1.** Sketch of a pressure ridge and its main dimensions depth, width and length.

While these models are based on mathematical relationships between different processes involved in ridging, DEM models
allows studying the detailed mechanics of the ridge formation process.

Earlier simulation-based studies on ridging have used two-dimensional DEM models. Hopkins et al. (1991) simulated ridging by compression of ice rubble consisting of circular and rectangular ice blocks between two floes. In further simulations by Hopkins (1994, 1998), a thin, intact, lead ice was pushed against a thick ice floe and went through a continuous ice failure process to form a ridge. Importantly, these simulations suggested a relation $F \propto h^{3/2}$ between the ridging force, $F$, and ice
thickness, $h$, which has been used since in some Earth System Models to define the strength of ice of a given thickness (Lipscomb et al., 2007).

Our study is the first to utilize three-dimensional DEM simulations to investigate ridging (Figure 2). In two-dimensional simulations, one cross-section of a ridge across its width is usually modelled, while three-dimensional studies can take the length of a ridge into account (Figure 1). First, we successfully validated our simulations by comparing our results to those
obtained experimentally by Tuhkuri and Lensu (2002). To study full-scale ridging processes, that is, ridging with ice thickness values typical to nature, we performed simulations with ice thicknesses and material parameters upscaled from laboratory scale to full scale. The upscaling used is based on Cauchy-Froude scaling, typically used in studies on ice-structure interaction (Schwarz, 1977; Timco, 1984). We show with these simulations that Cauchy-Froude scaling applies also for ridging. This finding gives confidence to our full-scale simulation results, opens an important avenue for future experimental work, and
sheds light on complex mechanics related to ridging processes. Further, we show that three-dimensional simulations indicate a relation $F \propto h$ between ridging force and ice thickness.

In what follows, we first briefly describe the model and introduce the setup for the simulations in laboratory- and full-scale in Section 2. Next, we present the results of the validation study and full-scale simulations in Section 3. We then discuss the results in Section 4 before concluding in Section 5.

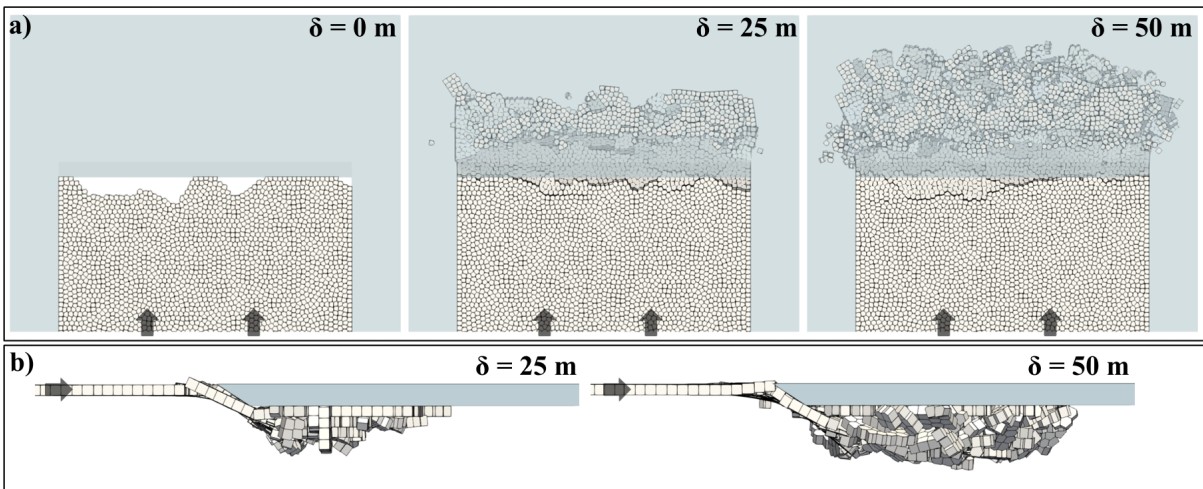

**Figure 2.** Snapshots from a DEM simulation of full-scale ridging viewed from the top (a) and side (b) at different distances of ice pushed $\delta$ into the ridge. Here 0.95 m thick ice (light gray) fails against a rigid ice floe (light blue). The ice moves in the direction of the rigid ice floe as indicated by the arrows. The simulation setup is described in detail in Section 2.2.

## 2 Methods

This section first describes our three-dimensional DEM model briefly. Then it explains the setups for ridging simulations in laboratory and full scale.

### 2.1 Numerical model

The numerical model employed is a three-dimensional discrete element method (DEM) code described in detail by Polojärvi (2022). The model was verified to describe the response and the fracture of an ice floe by Lilja et al. (2019a, b, 2021) and its results were successfully validated against laboratory-scale ice-structure interaction experiments by Polojärvi (2022). The model implementation is rather standard for DEM with features similar to those used in DEM modelling of sea ice since the 90s (Hopkins, 1992, 1994, 1998), as a central difference scheme is utilized for explicit time-stepping, rigid discrete particles interact through pairwise contacts, and deformation is described by using deformable finite elements connecting the rigid particles; models with the last feature are sometimes referred to as combined finite-discrete element models. The forces applied on the particles include internal forces due to ice deformation, contact forces and external forces due to gravity, buoyancy, and water drag.

In more detail, the deformable ice floe was simulated by employing a lattice of rigid, discrete particles interconnected by three-dimensional Timoshenko beam elements. The particles used are convex polyhedrons, generated through centroidal Voronoi tesselation (Du et al., 1999). The beam elements connect pairs of particles sharing a face. Both translational and rotational degrees of freedom of the beams correspond to those of the connected particles. The implementation of the beams

follows Crisfield (1990, 1997). Ice deformation and failure stem from the individual beams undergoing deformation and failure due to relative motion of the pair of particles connected by each beam. In the elastic regime, viscous material damping is used. Once the stress state of a beam meets a prescribed mixed-mode failure criterion (Schreyer et al., 2006), the beam undergoes a cohesive softening process, resulting in energy dissipation upon fracture (Paavilainen et al., 2009).

The model employs a soft-contact approach, wherein the contact force between a pair of interacting particles is determined based on a small particle-particle overlap volume, with the point of application of the force located at the centroid of this overlap volume. The contact force, $\mathbf{f} = \mathbf{f}_n + \mathbf{f}_t$, has a normal and a tangential component, $\mathbf{f}_n$ and $\mathbf{f}_t$, respectively. $\mathbf{f}_n$ is solved using an elastic-viscous-plastic contact force model by Hopkins (1992). The elastic and viscous portions of $\mathbf{f}_n$ are, respectively, calculated by using the gradient of overlap volume and its rate of change (Feng et al., 2012; Feng, 2021). The plastic portion of $\mathbf{f}_n$, describing local yielding at contacts, is solved based on contact area. Tangential compliance and friction between the particles contribute to the tangential force $\mathbf{f}_t$ (Hopkins, 1992). The contact model is parameterized by using the material properties of ice as described in Polojärvi (2022).

The model and its two-dimensional counterpart have been earlier used in studies on ice loads on inclined structures (Polojärvi, 2022; Paavilainen et al., 2009). These simulations indicate that $h$, which is in focus here, is a key parameter when defining ice load levels. The role of the other parameters in this is minor and, for example, the parameter controlling the plastic portion of $\mathbf{f}_n$ appears to merely affect the scatter in the peak ice load values (Ranta and Polojärvi, 2019).

## 2.2 Ridging simulations

This section first describes the setup of the simulations used to model the laboratory-scale experiments conducted by Tuhkuri and Lensu (2002). Then we describe how we upscale the setup and input parameters to full-scale by utilizing Cauchy-Froude scaling and perform simulations in full scale.

### 2.2.1 Laboratory-scale

We validated our simulations by comparing the modelled ridging force bluemagnitudes and ridge profiles to those measured in the laboratory-scale experiments by Tuhkuri and Lensu (2002). The experiments were conducted at the Aalto Ice and Wave Tank, an ice basin with an area of $40 \times 40\,\mathrm{m}$. In the experiments, 13 ice sheets were used to perform 38 experiments in total. In the experiments, each sheet was first cut into three $6\,\mathrm{m}$-wide strips with the surrounding ice left in place allowing to utilize the same ice sheet for three experiments. The strips were then cut in half and one of the floes was pushed against the other one. The horizontal force required to move the floe was recorded and defined as the ridging force, $F$, measured as the function of the distance the ice was pushed, $\delta$. From these three-dimensional ridging experiments, we chose four sets, S1 ... S4, of three experiments each, which all resulted in ridging. The material parameters, tensile strength, $\sigma_f$, and elastic modulus, $E$, and the ice thickness $h$ varied between these sets (Table 1). The total $\delta$ at the end of the experiments varied between $4.2\,\mathrm{m}$ and $12.0\,\mathrm{m}$. The ridges in sets S1 and S2 had their profiles measured at the end of the experiments at three equally spaced locations along the length of the ridge.

**Table 1.** Four sets S1 . . . S4 of parameterizations chosen after the laboratory-scale experiments by Tuhkuri and Lensu (2002). In the table, $h$ is the ice thickness, and $E$ and $\sigma_f$ are the elastic modulus, and tensile strength, respectively. Other main parameters are given in Table 2.

| set | $h$ [m] | $E$ [MPa] | $\sigma_f$ [kPa] |
|---|---|---|---|
| S1 | 0.095 | 27 | 10 |
| S2 | 0.089 | 24 | 16 |
| S3 | 0.078 | 64 | 12 |
| S4 | 0.048 | 368 | 37 |

The simulation setup featured a deformable ice floe moving at a constant velocity towards a rigid floe (Figure 2), from which we measured the sum of the horizontal contact forces, defined as the ridging force $F$ in the simulations. Similar to the experiments, additional ice floes on each side restricted the moving floe from lateral motion. Visual inspection showed that a 1-cm-wide gap between the moving floe and the floes on the sides was enough to avoid contact between them. The deformable floe had an uneven edge, while the rigid ice floe had a downward-sloping even edge at an angle of about $30°$ from the horizontal. These features were implemented ad hoc to avoid excessively high peaks in $F$ at the initial contact of the floes and to replicate the soft underside of the laboratory-scale ice. The thickness of the rigid floe was four times that of the deformable floe to avoid extensive rafting. Thus, the simulation setup was similar to the two-dimensional setup by Hopkins (1998), where an ice floe ridged against a $2.5 \ldots 5$ times thicker ice floe. The particles had an average aspect ratio of $1.5$ between the ice thickness and their width, defining the minimum aspect ratio for an ice block in our simulations. Consequently, the aspect ratio of the smallest ice fragments possible adhered to the observed lower limit of of $3.5 \pm 2.0$ for ridges in the Barents Sea (Høyland, 2007).

Similarly to the experiments, we performed four sets of simulations, S1 ... S4. While the parameters, which varied between S1 ... S4, were already given in Table 1, other simulation parameters are presented in Table 2. We repeated the simulations five times for each parameterization. In the repeated simulations, the tessellation and the shape of the edge of the deformable ice were varied, which were enough for the simulations to yield different failure processes as described in Polojärvi (2022). While in the experiments the ice was pushed with velocity of $0.01\,\mathrm{m\,s^{-1}}$, it was also shown that the ice velocity, at least in the range $0.01 \ldots 0.06\,\mathrm{m\,s^{-1}}$ tested, did not affect the results (Tuhkuri and Lensu, 2002). Thus, we used $0.05\,\mathrm{m\,s^{-1}}$ as the ice velocity in our simulations to cut down the wall-clock time required by our simulations. Finally, the simulated ice floes were of uniform thickness, while Tuhkuri and Lensu (2002) used ice of uneven thickness in their experiments. In two-dimensional simulations, Hopkins et al. (1999) simulated ridge formation due to compression of ice floes of similar, but nonuniform, ice thickness and concluded that this non-uniform thickness mainly influenced the ratio of rafting and ridging. We accounted for the effect of varying ice thickness on frictional sliding by using an ice-ice friction coefficient of $0.6$, which is at the high, but realistic, end for the values measured for ice (Sukhorukov and Løset, 2013).

**Table 2.** Main simulation parameters in the laboratory-scale simulations. Parameters are described in detail in Polojärvi (2022).

| Description | | Value | Unit |
|---|---|---|---|
| General | Time step | $2 \cdot 10^{-5}$ | s |
| | Gravitational acceleration | 9.81 | $\mathrm{m\,s^{-2}}$ |
| Ice | Floe width | 6 | m |
| | Floe length | 18 ... 25 | m |
| | Density | 930 | $\mathrm{kg\,m^{-3}}$ |
| | Velocity | 0.05 | $\mathrm{m\,s^{-1}}$ |
| Beams | Damping ratio | 0.75 | |
| | Shear strength | $\sigma_f$ | |
| | Poisson's ratio | 0.3 | |
| | Mean length | $1.5h$ | m |
| Contact | Plastic limit | 40 | kPa |
| | Ice-ice friction | 0.6 | |
| Water | Density | 1000 | $\mathrm{kg\,m^{-3}}$ |
| | Drag coefficient | 1.0 | |

### 2.2.2 Full-scale

To discuss the implications of the results below one must know how ridging processes scale; the relevant question to ask is if measured laboratory-scale (LS) ridging forces can be used to estimate those in full-scale (FS). Cauchy-Froude scaling, often used to scale experiments featuring ice-going ships and offshore structures, was also used here for the upscaling and later evaluated. In brief, with Cauchy-Froude scaling, geometric, kinematic and dynamic similitude are preserved (Schwarz, 1977; Timco, 1984). The Froude and Cauchy numbers are, respectively, given by

$$Fr = \frac{v}{\sqrt{gl}} \qquad \text{and} \qquad Ca = \frac{\rho v^2}{E}. \tag{1}$$

$Fr$ describes ratio of inertia to gravitational forces, represented, respectively, by velocity $v$ and gravitational acceleration $g$ and length scale $l$ applied to both horizontal and vertical dimensions. $Ca$, on the other hand, describes the ratio between inertia, represented by the density $\rho$ and velocity $v$, and elastic forces, represented by the elastic modulus $E$.

To test if Cauchy-Froude scaling is applicable, we first upscaled the parameters from LS to FS. Thus, all LS parameters (Table 1), including the ice thickness, material parameters and velocity, were upscaled using the scaling parameter $\lambda$:

$$h_{FS} = \lambda h_{LS}, \quad \sigma_{FS} = \lambda \sigma_{LS}, \quad E_{FS} = \lambda E_{LS}, \quad \text{and} \quad v_{FS} = \lambda^{1/2} v_{LS}. \tag{2}$$

The ice-ice friction was kept constant. If Cauchy-Froude scaling applies for ridging processes, then FS simulations should yield ridging forces matching those from LS simulations through scaling

$$F_{FS} = \lambda^3 F_{LS}. \tag{3}$$

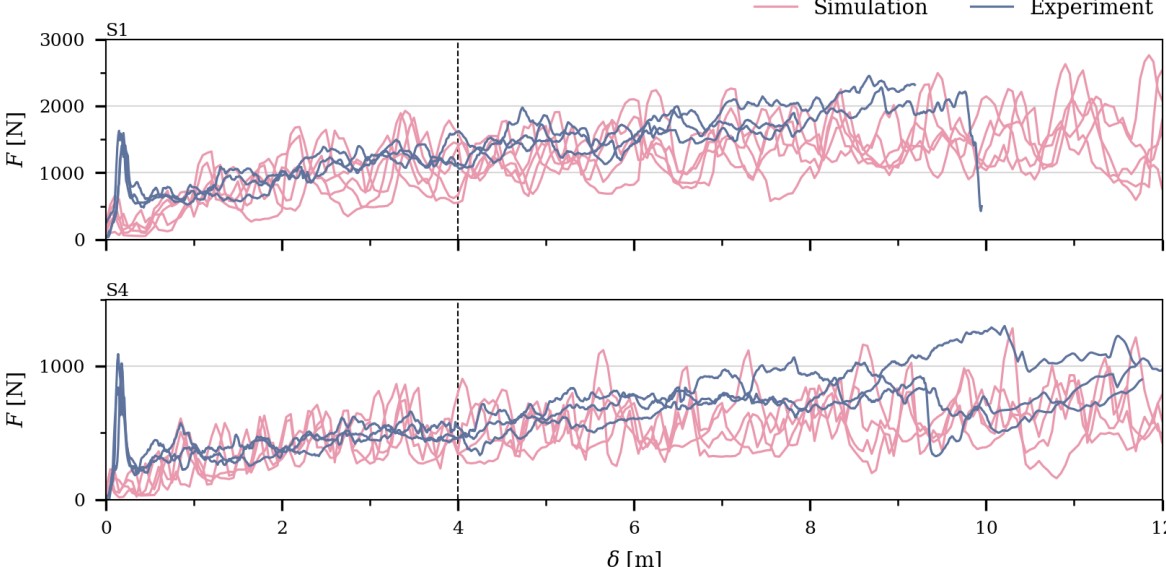

**Figure 3.** Ridging force, $F$, as a function of the distance of ice pushed, $\delta$, from the simulations and experiments with the thickest and thinnest ice (parameter sets S1 and S4, respectively). Each graph shows $F - \delta$ record from five laboratory-scale simulations and three experiments. The vertical line illustrates $\delta$ where the phase of the process changed from the initial phase to the second phase (Appendix A).

For the FS simulations here, we used $\lambda = 10$ based on Tuhkuri and Lensu (2002). We did not directly scale mean particle size, but instead used a mean aspect ratio of one for the particles in FS simulations, since preliminary simulations showed occasional sharp force peaks with larger particles. The peaks were likely due to the limited capability of the model to simulate local crushing and fragmentation of large particles (Polojärvi, 2022; Prasanna and Polojärvi, 2023). Through additional simulations we saw that the average ridging force was not affected by this choice. Further, the thickness of the rigid floe was adjusted to a maximum of $2\,\text{m}$. The thickness of the rigid floe was found to not influence the magnitude of the mean ridging force and only influenced the likelihood of rafting.

## 3 Results

We first show that the results from our simulations compare well with laboratory-scale experiments in regards to ridging forces and ridge geometries. Next, we use the simulations to show that Cauchy-Froude scaling applies to ice ridging.

### 3.1 Laboratory-scale

Figure 3 shows ridging force, $F$, plotted against the distance of ice pushed, $\delta$, in the simulations and the experiments. Only the $F - \delta$ records from the simulations and the experiments with the thickest ice and thinnest ice are shown, but these results

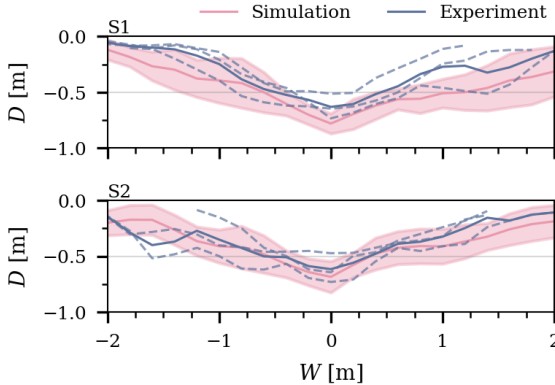

**Figure 4.** Comparison of mean ridge profiles (solid lines) from the experiments and simulations. All three ridge profiles per ridge for the experiments are displayed with dashed lines, while the standard deviation calculated from five simulated profiles is displayed as shading. The ridges were centered around their largest keel depth to ensure comparability.

are representative for all data. Excluding the initial peak in the experimental $F - \delta$ records, the ridging-force levels from the experiments and simulations are in good agreement, which already suggests that our simulations describes ridging well and thus partly validates our approach. The initial peak in experimental $F - \delta$ records is due to the two ice floes with even edges colliding and it is not expected to be present in data from the simulations (Section 2.2.1). It should be emphasized that the

initial peak $F$ should not be treated as a ridging force, since it does not represent the force required to build a ridge.

Following Hopkins (1998), the $F - \delta$ records were divided into an initial phase, during which $F$ increased with $\delta$, followed by a second phase of $F$ fluctuating about a constant mean (Appendix A). During the initial phase, $F$ increased with an approximately equal rate in the simulations and the experiments. The first phase continued up to $\delta \approx 4\,\mathrm{m}$ indicated by the dashed line in Figure 3. In general, the change from the initial to second phase is more distinct in the simulations than in the experiments.

During the second phase, $F$ was then estimated to fluctuate around a mean force, $\bar{F}_{II}$. Magnitude of $\bar{F}_{II}$ in the simulations and the experiments compared well: $\bar{F}_{II}$ from the simulations was within one standard deviation of that in the experiments.

In addition to $F - \delta$ records, Tuhkuri and Lensu (2002) provided ridge profile measurements, which we compare to mean ridge profiles from the simulations in Figure 4. The profiles from the end of each experiment were available for ridges in sets S1 and S2. For profiles yielding from the simulations at equal $\delta$, the thickness of the rigid floe is subtracted from the entire

profile. Overall, the mean ridge profiles from the simulations are in agreement with the profiles from the experiments in shape and depth (Figure 4). Some ridges in the experiments are shallower than the mean profile from the simulations. This difference is expected, as the ice in the experiments can crush and fragment into very small pieces, leading to potentially more compacted ice rubble than in the simulations, where the smallest ice fragment size is governed by the particle size. Nevertheless, the majority of the ridges from the experiments have their profiles within one standard deviation from simulation profiles.

Additionally, we analyzed how depth and width, $D$ and $W$, respectively, of the ridge developed as function of $\delta$. $D$ was defined based on the deepest point of the keel with the thickness of the rigid floe subtracted from it. $W$ was defined based

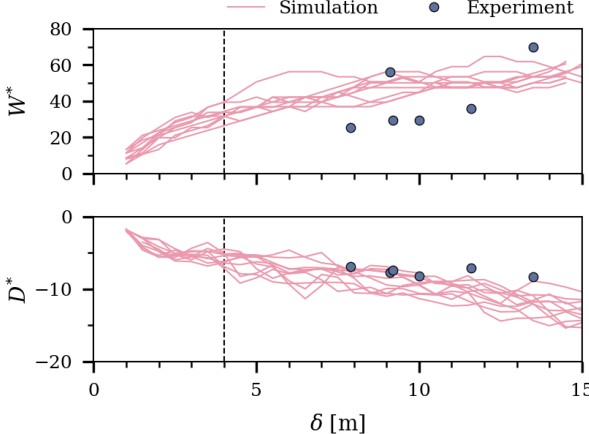

**Figure 5.** Evolution of the normalized ridge width $W^* = W/h$ and depth $D^* = D/h$ from S1 and S2 laboratory-scale simulations. Data are normalized by ice thickness $h$ and plotted against the distance of ice pushed $\delta$. Experimental data were only available for the experiments with parameter sets S1 and S2. Vertical dashed line represents the change from the initial phase to the second phase (Appendix A).

on the depth of rubble with $D > 2h$. Figure 5 illustrates how $D$ and $W$ developed for the S1 and S2 sets by showing their thickness-normalized values $W^* = W/h$ and $D^* = D/h$, which accounts for the various values of $h$ used in the simulations. It can be seen that $D^*$ increases throughout the whole simulated process, while $W^*$ first showed a steady increase and then
periods of constant or slowly increasing $W^*$. Figure 5 also shows experimental data points (S1 and S2) based on the profile measurements at the end of the experiments. Data from the simulations and the experiments can be seen to be in fair agreement.

In addition to the comparison of ridging forces and geometry, the simulations allow observing ridging processes in detail. Each ridging simulation started with the deformable floe approaching the rigid floe, followed by pieces of the uneven edge breaking off. After the initial interaction, the floe started to submerge and break into rubble, which generally accumulated
in one layer and could be related to initial rafting. Then, the intact, deformable floe continued to bend downwards and fail, either against the rigid floe or the accumulated ice rubble. After the new ice rubble was created, it moved further under the rigid floe with the ridge growing in depth and width (Figure 5). During this process, the rubble pieces can break and rearrange. Nevertheless, the ice rarely broke into the smallest possible particle size, resulting in rubble pieces consisting of several connected particles. From observations of several ridging simulations, we identified two main deformation processes
during ridging: first, creating more ice rubble to be added into the ridge and, second, further deforming and transporting the ice rubble within the ridge.

## 3.2 Full-scale

Next, we demonstrate that Cauchy-Froude scaling is applicable to ridging. To do this, Figure 6 compares the $F - \delta$ records from FS simulations to those from LS simulations upscaled with Cauchy-Froude scaling to FS (Equation 3). This comparison

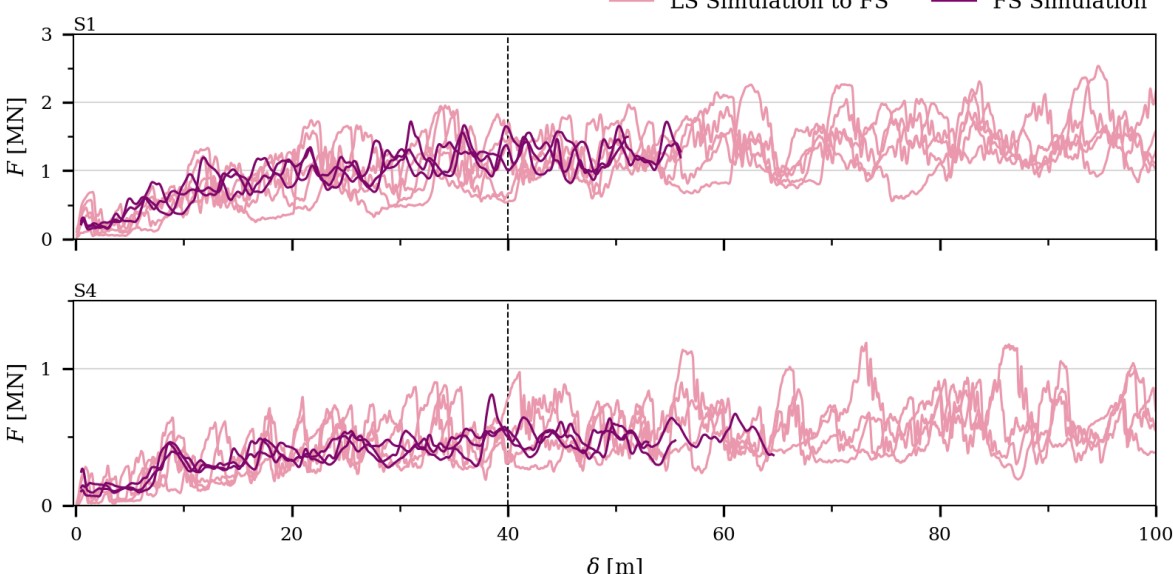

**Figure 6.** Ridging force, $F$, as a function of the distance of ice pushed, $\delta$, from full-scale (FS) simulations and laboratory-scale (LS) simulations upscaled with Cauchy-Froude scaling (Equation 3) to FS for the thickest and thinnest ice (parameter sets S1 and S4, respectively). Each graph shows $F - \delta$ records from five LS simulations and three FS simulations. The $F - \delta$ records are divided into two phases, with the change from the initial phase to the second phase illustrated by the dashed line (Appendix A).

shows that the overall features of the $F - \delta$ records are similar and the force levels in the simulations conducted at different scales are in agreement.

The $F - \delta$ records of Figure 6 was again divided into two phases (Appendix A), with the second phase of $\bar{F}_{II}$ starting at $\delta \approx 40\,\mathrm{m}$. This pattern would be expected if Cauchy-Froude scaling applied for ridging, since the second phase started $\delta \approx 4\,\mathrm{m}$ in LS simulations and scaling factor $\lambda = 10$ was used. The mean $F$ during the second phase, $\bar{F}_{II}$, from the simulations 190 performed in different scales matched well (Figure A1).

In addition to similar $F - \delta$ records, $W$ and $D$ of FS ridges evolved similarly to LS ridges. Since the main features of ridge geometry were similar in both scales, and the $F - \delta$ records matched with the chosen scaling, we conclude that the model can be applied on both scales and that Cauchy-Froude scaling applies for the simulated ridging process and may, thus, apply for ridging in nature as well.

## 4  Discussion

Formation of pressure ridges has been simulated earlier only by utilizing two-dimensional DEM models (Hopkins, 1994, 1998; Hopkins et al., 1999; Damsgaard et al., 2021). Since simulating pressure ridging in three-dimensions is a fairly complex effort, it

is relevant to discuss how our results differ from two-dimensional simulations. One important feature only a three-dimensional simulation allows is the non-simultaneous failure process. Non-simultaneous failure is a well-known feature related to sea ice interacting with offshore structures (Ashby and Hallam, 1986; Sanderson, 1988). In this case, the ice floe acting on the structure does not fail uniformly across the whole length of its nominal contact area, defined as $hL$, but only at distinct locations at any given time instant. In keeping with this analogy, the ice failure in our full-scale simulations never occurred across the whole length of the ridge at once, but rather through seemingly independent smaller failure processes and failure events at distinct locations across the length of the ridge (Figure 2).

A non-simultaneous failure process cannot be described by two-dimensional simulations. Absence of this feature manifests, for example, as abrupt drops in the $F - \delta$ records to zero. These force drops occur when the contact between the interacting ice floes is momentarily lost upon ice failure as seen from results by Hopkins (1998). Our $F - \delta$ records do not show such force drops, but we ran supplementary simulations with shorter ridges. These simulations confirmed that with decreasing $L$, the fluctuations in the $F - \delta$ records became more pronounced. Comparing $F - \delta$ from simulations with $h = 0.95\,\mathrm{m}$ and $L = 10\,\mathrm{m}$ to simulations with $L = 60\,\mathrm{m}$ resulted in an about 90 % higher standard deviation for $\bar{F}_{II}$ together with periods of virtually zero $F$ (Muchow and Polojärvi, 2024). Non-simultaneous failure, allowed by three-dimensional simulations, is required for realistic modelling of ridge formation and estimates of ridging forces.

Tuhkuri and Lensu (2002) observed that in their experiments ridging started with rafting. Based on this finding, they suggested a simplified ridging model including two phases, the first phase with an increasing $F$ due to rafting and the second phase with a constant $F$. This concept is supported by dissection of full-scale ridges, which featured several layers of rafted ice close to the waterline as a consolidated layer (Høyland, 2007). This observation made them suggest that rafting is also part of full-scale ridging. Our initial simulations in laboratory-scale showed that our model is prone to yield very extensive rafting, leading us to develop and use the setup as described in section 2.1. This setup is similar to the two-dimensional setup of Hopkins (1998) with thin lead ice ridging against thicker lead ice. The simulations performed here do not show an initial phase of pure rafting as the ice fails into discrete ice blocks while being submerged. Nevertheless, the later ridge formation process is still described well and the $F - \delta$ records from the simulations match well with those from the experiments. Further, when accounting for the fact that we set up our simulations so that the initial peak force in $F - \delta$ records was removed, it appears that the ridging process and magnitude of $\bar{F}_{II}$ are apparently not affected by the details of the initial stages of pressure ridge formation.

What do our simulations then tell about the mechanisms that limit the magnitude of $F$? This is an important question related to the development of rheological models for pack ice and estimating the large-scale ice strength, which is assumed to be limited by the ice failure processes such as pressure ridge formation (Tuhkuri and Lensu, 2002; Lipscomb et al., 2007). Figure 7 aims to answer this question by plotting the values of average ridging force per unit length, $\bar{F}_{II}L^{-1}$, against $h$ for all simulations and experiments described above. (Values for $\bar{F}_{II}L^{-1}$ from LS simulations and experiments presented were scaled to FS according to Equation 3.) Figure 7 also shows $\bar{F}_{II}L^{-1}$ for additional FS simulations with $h > 1.0\,\mathrm{m}$, ran with FS parameterizations of S1 an S4 and preserved $L/h$ ratio of S1 (Table 1). In addition to the data points, Figure 7 presents a linear fit for the full-scale data suggesting $\bar{F}_{II} \propto h$. The linear fit has a Pearson correlation coefficient of 0.99, which indicates a nearly perfect fit. This result

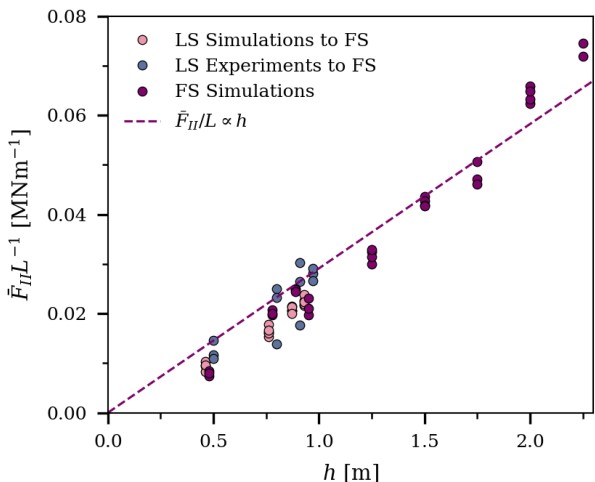

**Figure 7.** Mean ridging force, $\bar{F}_{II}$, during the second phase ($\delta > 40\,\mathrm{m}$) divided by the ridge length $L$ for each individual FS simulation is shown depending on the ice thickness $h$ of the deformable ice. Simulations with $h > 1.0$ m are additional simulations conducted with thicker ice with S1 and S4 material properties (Table 1). The dashed line shows a linear fit applied to the FS data. The laboratory-scale (LS) experiments and simulations are both upscaled to FS (Equation 3), and only included as additional information.

differs from $F \propto h^{3/2}$ found earlier when using two-dimensional DEM simulations (Hopkins, 1998; Damsgaard et al., 2021). Higher degree polynomials typically lead to better fits, and for our data, a fit for $F \propto h^{3/2}$ would yield correlation coefficient with a difference in the third decimal place; we argue for $F \propto h$ as it is a first degree polynomial. We repeated the analysis by using the maximum force magnitudes, which also ended up showing a clear linear dependency on $h$. Analogue to the inclusion of $L$ into ridging simulations, two- and three-dimensional DEM simulations on ice-structure interaction processes appear to yield a similar behavior. Two-dimensional simulations on ice loading on an inclined structure suggested that the ice load is proportional to $h^{3/2}$, while three-dimensional simulations of ice acting on a narrow upwards-bending conical structure yield a linear relationship between ice load and $h$ (Ranta et al., 2018; Polojärvi, 2022). Combining these observations, it appears even more crucial to simulate ridging in three dimensions; three dimensional DEM suggests ice strength, as often defined in Earth System models, should increase linearly with $h$.

From the aspect of improving large-scale compressive ice strength estimates, our results have the following interesting implication. During full-scale simulations, it appears that a $L/h$ ratio of 30, here nearly 1-meter-thick-ice forming a 30-meter-long-ridge, is enough to account for the effect of non-simultaneous failure. This ratio suggests that full-scale experiments on ridge formation processes could be performed to further confirm the results presented here and yielded by earlier laboratory-scale work. It is, however, beyond the scope of this paper to start numerically deriving the exact combinations of parameters applicable for executing such experiments.

## 5 Conclusions

This is the first study to use a three-dimensional discrete element method (DEM) to study pressure ridging. The model used was originally presented in Polojärvi (2022). Simulations were first ran by using laboratory-scale parametrization, which were then upscaled to full scale by using Cauchy-Froude scaling. Based on the simulation results, we conclude:

– The numerical model was successfully validated by comparing the ridging force records and ridge geometries to laboratory-scale experimental data (Figure 3 and 4).

– Cauchy-Froude scaling is applicable to the simulated ridging process (Figure 6). This result opens new avenues for experimental studies and gives new insight on mechanics of ridging processes.

– Three-dimensional simulations also account for the ridge length, which facilitates non-simultaneous failure. This expansion is important as it allows modelling realistic ridging processes.

– There is a linear relationship, $F \propto h$, between the ridging force and ice thickness (Figure 7). It seems that this finding can only be reached with three-dimensional simulations.

The ice thickness is likely the key parameter related to ridge formation process, yet for further insight, it would be beneficial to perform a detailed study on the effect of additional ice parameters on ridging in the future. The next steps for numerical studies on ridging should also include simulations with an enlarged sea-ice area so that several ridges may form and to account for scenarios, where ice floes of different thicknesses interacts with each other. These simulations would yield crucial information for the large-scale ice strength and answer open questions about how ridging influences sea-ice redistribution as used in Earth System Models.

## Appendix A:  Two phases of the ridge formation process

We divided the ridge formation process into two phases following Hopkins (1998) and a conceptual model by Tuhkuri and Lensu (2002). The first phase is characterized by an increase in force $F$, while the second phase shows $F$ fluctuating around an nearly constant average $F$. Here, the phases were identified by using the $F - \delta$ records and the rate of change their slope as follows. First, a running mean with a window of $\delta = 10\,\mathrm{m}$ was applied to the mean $F - \delta$ records, defined for each simulation set. The resulting force record was then used to calculate the slope $\Delta F / \Delta \delta$ for each $20\,\mathrm{m}$ meter interval. The $F - \delta$ mean as well $\Delta F / \Delta \delta$ are shown in Figure A1. First phase was defined to end at $\delta = 40\,\mathrm{m}$, where $\Delta F / \Delta \delta$ becomes negligibly small and the second phase starts. The average force during the second phase, $\bar{F}_{II}$, was used to analyze the relationship between the ridging force and ice thickness (Figure 7).

*Author contributions.*  MM set up the simulations, performed them, analyzed the results, and wrote the paper. AP implemented the DEM code, and contributed to the analysis and writing of the paper.

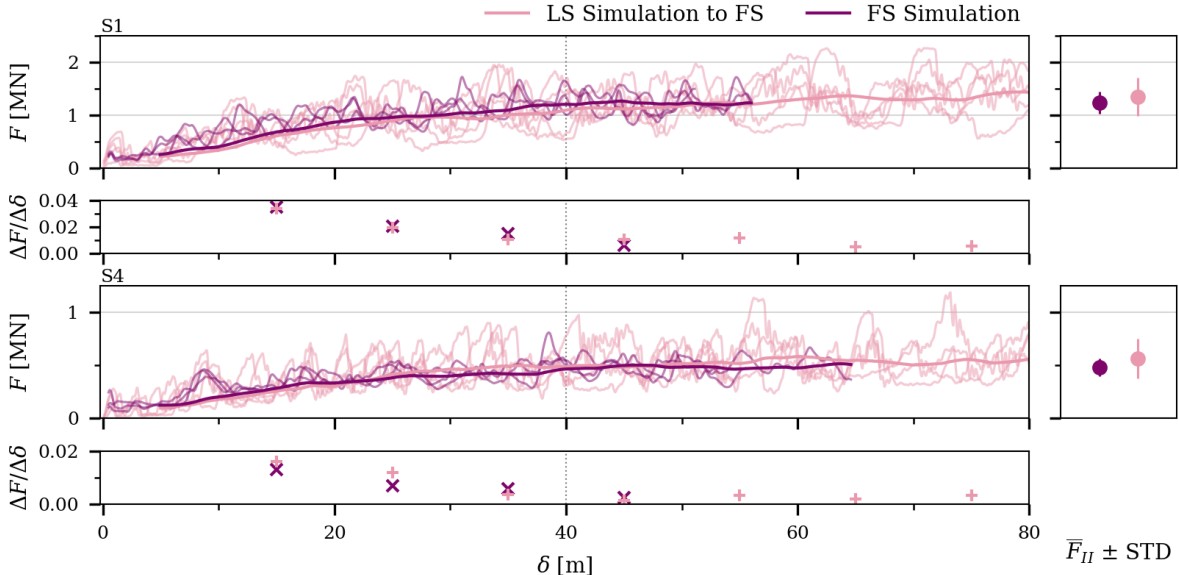

**Figure A1.** Ridging force, $F$, as a function of the distance of ice pushed, $\delta$, from full-scale (FS) and laboratory-scale (LS) simulations, latter upscaled to FS (Equation 3) for simulations in sets S1 and S4 (Tables 2). First and third graph show $F - \delta$ records from five LS simulations and three FS simulations and their running mean, while second and forth graph show the slope $\Delta F/\Delta \delta$, defined as decribed in Appendix A. Change between the first and second phase of the ridging process is illustrated with the dashed line. The mean force during the second phase, $\bar{F}_{II} \pm$ and its standard deviation (STD) is given in the right column.

*Competing interests.* The authors do not have competing interests.

*Acknowledgements.* The authors wish to acknowledge funding from the European Union – NextGenerationEU instrument through Research Council of Finland under grant number (348586) WindySea – Modelling engine to design, assess environmental impacts, and operate wind farms for ice-covered waters. CSC – IT Center for Science (Finland) is also acknowledged for computational resources. MM acknowledges funding from the AaltoENG 4-year doctoral program. Additionally, we thank Jukka Tuhkuri and Mikko Lensu for providing us with the experimental data. We sincerely thank the editor David Schroeder and two anonymous reviewers for their time, constructive feedback and insightful comments.

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
