# Peer review of "Three-dimensional discrete element simulations on pressure ridge formation"

_EGUsphere, 2024_

## Referee Comment (RC2)

**Review: Three-dimensional discrete element simulations on pressure ridge formation**

June 2024

**1 General Comments**

In this paper, the authors develop 3D DEM simulations to model the formation of sea ice ridges. Although the methodologies are not novel, the application to the full 3D scenario is new and the results are potentially significant. The authors illustrate the necessity for all three dimensions and non-simultaneous failure in order to accurately capture the phenomenon, and is therefore, in my opinion, an advancement of the state of the art for simulating pressure ridges. The authors also argue that their simulations result in a linear relationship between ridging force and ice thickness ($F \propto h$), which differs from previous relationships in the literature ($F \propto h^{3/2}$). However, this linear relationship appears analogous to recent publications of ice crushing against solid structures, as noted by the authors. Despite this, this result would be significant as the previous ridging relationships have been integrated into ESMs for decades. I have asked some questions below to clarify aspects of this linear fit. Overall, the document is well written, interesting, and presented in an accessible way.

**2 Specific Comments**

1. Lines 25-30 - It might be beneficial to provide some background information on the Hopkins model such as what particle geometry they used (polygons vs disks), how they handled contact mechanics, how they modeled inter-particle bonds (and their failure), etc. This would give the reader some context as to what the previous state of the art was versus your approach.

2. Lines 67-68 - I understand that the plastic portion of $f_n$ approximates local yielding and crushing. Do you have any comments on the relationship between this local deformation parameter and the large scale deformation ridging process? Did you do any analysis of how the magnitude of this plastic portion affects the ridging results? Or have any thoughts on how it may relate to the larger scale deformation?

3. Lines 86-86 - Just confirming I understand this - is this saying that the ridging force was measured as the contact force on the rigid floe?

4. Line 88 - A 1 cm gap seems small for a domain spanning several meters. What kind of analysis/measurements were done to make sure there were zero frictional forces from the adjacent ice throughout the simulation?

5. Lines 92-94 - The discussion related to the particle aspect ratio tripped me up a couple times. I believe you are saying that a particle aspect ratio of 1.5 resulted in simulated ridges that matched the aspect ratio of ridges measured in Høyland, 2007. Is that a correct interpretation? Was the 1.5 value determined by iterating through different particle aspect ratios? Can you make any comments about the effect that particle aspect ratio has on the resultant ridge geometry?

6. Lines 101-102 - Do you have any comments on how variable thickness may affect the simulated results, or how they might contribute to differences between the model and experiment?

7. Equations 2 and 3 - Can you briefly explain why the scaling parameters have different exponents for the velocity term in Equation 2 and the force term in Equation 3?

8. Lines 138-143 - Both the simulated and experimental data seem like they are constantly increasing. Are there statistical tests that could show evidence of the two phases? Perhaps compute a moving window average and evaluate its slope in each phase? Or compute a regression line for each phase, and then compare the slopes of each? Along the same idea - you mention that the change from first to second phase is more pronounced in the simulations than in the experiments - do you have any comments or thoughts on why that is the case?

9. It's not exactly clear to me why splitting the data into two phases is significant. Is the main idea that *if* the ridging force is more or less constant in the second phase, and that you can then use that to formulate some $F \propto h$ law? Assuming you do not need to split into two phases, could you use the maximum/peak ridging force instead of the mean force in the data fit?

10. Line 145 - How were the simulated ridge profiles computed? Referencing Figure 2b, the bottom surfaces look more "bumpy" than the profiles in Figure 4. Were the bottom-most particle positions sampled at some sort of regular interval along the width?

11. Line 156 - $W^*$ appears to continuously increase, and does not "plateau."

12. Line 196 - What is the "higher" in comparison to? 90% higher std dev than what simulation case?

13. Line 203 - Can you explain a little more what is meant by 'the setup as described above"? Do you mean your general simulation geometry? Or the general 3D DEM approach? Something else?

14. Lines 214-215 - Did you try to fit a $F_{II} \propto h^{3/2}$ type formula to this data? I would be curious to know what the Pearson coefficient is for that kind of fit. If you are arguing that a linear fit is more appropriate, then it makes sense to show the correlation coefficient of that fit, too, for comparison. The tail end of the data in Figure 7 appear to trend above the linear fit - did you run any simulations with thicker ice? It may be interesting to see if the $F \propto h$ relationship holds as $h$ increases.

15. Lines 218-219 - You reference your 2022 study that showed a linear relationship between ice load and thickness for simulations of ice against a rigid cone structure. Can you comment on the novelty of finding a similar relationship in this current manuscript? The larger floe in this paper was also modeled as a rigid structure, so the simulation setup seems fairly similar to the 2022 paper.

**3   Grammar/Spelling Corrections**

Suggested corrections are indicated in **bold**.

1. Line 1 - "...discrete element method simulations **of** pressure ridge formation."

2. Line 21 - "**The** first theoretical models for..."

3. Lines 76-77 - "We validated our simulations by comparing **the modeled** ridging force **magnitudes** and ridge profiles to those **measured in** the laboratory-scale experiments by Tuhkuri and Lensu (2002)."

---

## Author Comment (AC1)

*We sincerely thank the reviewers for their insightful comments and constructive feedback. Their suggestions have significantly improved the quality of our manuscript. We appreciate their time and effort in reviewing our work. Below, all answers are written in cursive blue next to the original comments in black.*

**Answers: Reviewer 1**

**General Comments**

Thanks for the opportunity. I enjoyed reading this manuscript.

This paper describes simulations conducted with a three-dimensional DEM-FEM sea ice model to study the dynamics of ridging. The study is grounded by a comparison of the model performance at laboratory scale with an experiment performed by Tuhkuri and Lensu (2002). The model simulations are then extended to a larger scale to see how the process scales. The authors draw conclusions about how the ridge force should scale with ice thickness in a three-dimensional setting, and about how ridge behavior might follow Cauchy-Froude scaling across scales. While I think the authors need to add some details, clarify some assumptions and couch some conclusions more carefully, their methodology is sound and the manuscript is well written and organized. Their conclusions are largely justified, in some ways provocative and worthy of publication.

**Science related questions** (simply in the order in which they occured to me in reading the manuscript)

I think this experiment applies to unconsolidated ridges. If that is the case, it is worth noting somewhere near the top. *- Correct. We added a sentence in the first paragraph of the introduction.*

Line 35: Definition of Full-scale. My understanding is that ridges form over a variety of scales. So, I am not clear on what 'full scale' specifically means here. A reference would be useful here, if there is observational evidence for a particular 'scale of a natural ice ridge'. From reading, it seems that for this manuscript's purposes full-scale is 10 times laboratory scale. But this should be stated upfront with some justification for the choice. *- Here 'full scale' ridges are assumed to be representative of ridges formed from typical level ice thicknesses in nature. Thus, we refer to the scaling of the thickness of level ice. Tuhkuri and Lensu (2002) chose a scaling factor 10 for their laboratory scale experiments, thus we used that factor. This information is added to the respective line.*

Line 51: DEM or FEM-DEM? I would not describe the FEM-DEM method being used here as 'rather standard', the beam component is more sophisticated than most that I am familiar with. Throughout the manuscript, the model is called a DEM. But I am not sure if DEM with FEM joints is referred to elsewhere in the literature as DEM. *- This is now addressed in Section 2. The terminology for particle-based techniques remains imprecise, with DEM, FEM-DEM, and the bonded particle method (BPM) often used for similar approaches. We opted for DEM but now clarify that term FEM-DEM is also used for models with embedded finite elements.*

Line 84: Do the authors think that the Tuhkuri and Lensu experiments captured 3-dimensional aspects of ridge formation? From the arguments later in the paper regarding the ridge force scaling linearly with h, it seems that 3-d behavior is already exhibited at the lab scale, if it is 3d behavior that leads to linear scaling of the force with h. *- Yes, we think that the three-dimensional behavior is already shown in the laboratory-scale experiments, but on a smaller range of ice thicknesses which complicates the analysis of the force-thickness relationship. We added the description 'three-dimensional' to the experiments.*

Line 93: It would be useful if the authors would comment on the implication of having one side of the ridge, being a thick unbreakable sheet as compared to two breakable sheets of equal thickness. I understand that with equal thicknesses the simulated ice rafted too much. I would like to know why that was the case. But also, this setup seems like an ice-structure interaction rather than a ridge formation between sheets of ice. For example, this setup does not allow a sail to form in addition to a keel. I imagine this might be a fair idealization as a first shot of handling this problem in 3D. I would just like to hear more justification for the setup. *- The set up is now similar to Hopkins (1998), where thinner lead ice ridges against a thick ice floe. We do not yet have a clear answer to the question related to extensive rafting. During development, we tested, for example, uneven thickness distribution across the floe as it was featured in the experiments to facilitate ridging. This proofed not to be as fruitful as hoped as it introduced more variables into the analysis of the results, and introduced artificial weak points and made our ice fail in an artificial manner.*

Line 94: Can this be read to mean that you chose the aspect ratio of your elements to get simulated ice blocks in the rubble of the ridge that had the right characteristics compared to field observations? *- We reworded the sentence in question. We chose the aspect ratio so that it was possible for the ice to fail into blocks that could have the aspect ratios observed in the field. Discrete element size (defining the smallest fragment size) was, thus, chosen so that pieces could reach aspect ratios down to 1.5 observed in the field (Høyland, 2007).*

Line 100: Please explain why an ice velocity of 0.05 m s-1 is used rather than 0.01 m s-1 Was it for computational efficiency or due to other considerations? *- Yes, the velocity was chosen to cut down the wall-clock time of our simulations and is added as reasoning to the respective sentence.*

Line 112: Please clarify if 'l' in the Froude number is meant to be a vertical length scale, horizontal, or if they're considered comparable. *- The length scale l refers to horizontal and vertical length scales likewise. We added this for clarity.*

Line 126: From the earlier statements one would conclude that the thickness of the rigid floe affects the ridging force. In the LS it was chosen to be larger than the moving floe, because there was excessive rafting if it were not. So, this statement seems in conflict with other statements, unless I'm not interpreting it correctly. Intuitively it seems like the 'l' in the Froude number might be related to the thickness of the unbreakable sheet. *- The thickness of the rigid floe does not influence the forces, it just influences the likelihood of the ice to ridge or raft. We added this to the mentioned sentence.*

Line 142: I would not say that 'F fluctuates around a mean value' in the second phase of the

experiment. To my eye it increases linearly in both the experiment and in the simulations (at a lesser slope). Given this, I'm not sure how the standard deviation is calculated for the figures, and if it is representative. By eye it looks like the simulations have notably greater standard deviation than the experiments unless you include the linear trend as part of the std. *- Based on the other reviewers suggestion, we looked into the distinguishing of the phases via applying running means to the data and then comparing the slopes of these running means. For the simulations, this analysis shows a clear change in slope and thus, we argue that the splitting in stages is justified. For the experimental data, we decided to not split the data in stages. Thus, Figure 3 changes in the manuscript and we add additional analysis of the stages.*

Line 156: The plateau in W* is not visible to me. *- We agree that it is not a constant plateau, but rather periods where ridges do not grow in width. We adapted the text accordingly.*

Line 180. I think what has been demonstrated is that the model follows Cauchy-Froude scaling, but this does not necessarily mean that nature does. This conclusion should be stated a bit more carefully. *- We agree and adapted the mentioned section.*

Line 191-197. I think this is an interesting point, potentially deserving of a figure if the supplementary simulations could supply one. *- We added one additional figure to add to the discussion of non-simultaneous failure. The new figure shows snapshots of two simulations with different ridge length L to highlight the non-simultaneous failure.*

Line 222-225. Related to my previous (Line 191) comment. I do not fully understand the basis for the statement starting on line 225. I take it to mean that the ridge length was varied in FS experiments, and at a 30:1 L/h ratio, non-simultaneous failure behavior was observed. But would the non-simultaneous failure behavior differ at 100:1 or 1000:1? The author's again use the term full-scale here, but might not things change at floe scales larger than the 'full-scale' considered here. For that matter the simulated LS scale gave results quite similar to the FS, which returns to my earlier comment (Line 84). It would be good if the authors could disentangle their conclusions a bit better here with regards to scale and 3d vs 2d behavior. *- The ratio of 30:1 is the minimum we suggest to achieve non-simultaneous failure. We assume, that there is no upper limit for this behaviour. Additionally, our simulations are not large-enough (in spatial dimensions) to argue for a maximum limit. We can also not exclude that potentially other processes limit ridging forces on larger length scales. The last paragraph of the discussion is intended to highlight the possibility of experiments in large ice tank facilities, without the need of scaling the parameters to laboratory scale. We hope that with the clarification of what full scale refers to and some rewording, this statement also becomes more clearer.*

**Technical corrections**

Line 3: suggestion: 'at' large scale *- Changed.*

Line 6: suggestion: 'along the contact interface' rather than 'across the ridge length'. *- We appreciate the suggestion, but will continue to use ridge length as it is used throughout the manuscript.*

Line 13: no need for 'Thus' *- 'Thus' is removed.*

Line 21: 'The' first *- Changed.*

Line 24: ambiguous phrase. What are the different processes involved and how do they differ from 'the process itself'? *- Different processes are, for example, the failure of the intact ice floe and the redistribution of ice. With 'the process itself' we tried to highlight, that we explicitly simulate ridging. We adapted the sentence.*

Line 81 possibly replace 'of the amount of ice pushed', with 'the distance the ice was pushed'? *- Changed in this and all other locations.*

Line 82: Were the 3 experiments in each of the four sets (S1..S4), replicates of each other? *- Yes, all three experiments within each set used the same ice sheet and thus, have similar material properties. We added some words.*

Line 145: At what point in the experiment and simulation are the profiles taken delta=4m? 10m? Line 158 suggests it's 'at the end of the experiment' but should probably say that here. *- The ridge profiles from the experiments are measured at the end of each experiment. We agree and added context.*

Line 173: 'conducted at different scales' *- Changed.*

Figure 6, etc. The terminology 'LS simulations to FS' isn't explained anywhere. Although I'm guessing it means that the LS variables are CF scaled (Force is multiplied by $\lambda^3$ for example). *- Yes, this is what we mean with 'LS simulations to FS'. We changed the caption of all Figures where this description appears.*

---

## Author Comment (AC2)

*We sincerely thank the reviewers for their insightful comments and constructive feedback. Their suggestions have significantly improved the quality of our manuscript. We appreciate their time and effort in reviewing our work. Below, all answers are written in cursive blue next to the original comments in black.*

**Answers: Reviewer 2**

**General Comments**

In this paper, the authors develop 3D DEM simulations to model the formation of sea ice ridges. Although the methodologies are not novel, the application to the full 3D scenario is new and the results are potentially significant. The authors illustrate the necessity for all three dimensions and non-simultaneous failure in order to accurately capture the phenomenon, and is therefore, in my opinion, an advancement of the state of the art for simulating pressure ridges. The authors also argue that their simulations result in a linear relationship between ridging force and ice thickness (F h), which differs from previous relationships in the literature (F h3/2). However, this linear relationship appears analogous to recent publications of ice crushing against solid structures, as noted by the authors. Despite this, this result would be significant as the previous ridging relationships have been integrated into ESMs for decades. I have asked some questions below to clarify aspects of this linear fit. Overall, the document is well written, interesting, and presented in an accessible way.

**Specific Comments**

1. Lines 25-30 - It might be beneficial to provide some background information on the Hopkins model such as what particle geometry they used (polygons vs disks), how they handled contact mechanics, how they modeled inter-particle bonds (and their failure), etc. This would give the reader some context as to what the previous state of the art was versus your approach. *- We added information on the particle shapes into the mentioned paragraph. Additionally, we now included into Section 2.1 that the general features of our model are similar to those used in sea ice DEM since 90s.*

2. Lines 67-68 - I understand that the plastic portion of fn approximates local yielding and crushing. Do you have any comments on the relationship between this local deformation parameter and the large scale deformation ridging process? Did you do any analysis of how the magnitude of this plastic portion affects the ridging results? Or have any thoughts on how it may relate to the larger scale deformation? *- This is an interesting point with related changes now in Section 2.1 and at the end of the conclusions. We describe that ice thickness has been found to be a key parameter affecting ice loads. 2D simulations in Ranta (2019) suggest plastic limit explains scatter in the values of peak ice load on inclined structures, yet does not have an effect on load levels. We also now mention that a future detailed study on parameter effects would be beneficial.*

3. Lines 86-86 - Just confirming I understand this - is this saying that the ridging force was measured as the contact force on the rigid floe? *- Yes, we add 'sum of contact forces' to the*

*sentence.*

4. Line 88 - A 1 cm gap seems small for a domain spanning several meters. What kind of analysis/measurements were done to make sure there were zero frictional forces from the adjacent ice throughout the simulation? *- When developing the setup, we checked visually that the approaching ice was not touching the ice acting as side restrictions. As the direction of the ice velocity is parallel to the adjacent ice, contact did not occur. If contact occurs closer to the ridge, it would still not interfere with the recording of the ridging forces as the forces on the side restrictions are not included in the overall force recordings.*

5. Lines 92-94 - The discussion related to the particle aspect ratio tripped me up a couple times. I believe you are saying that a particle aspect ratio of 1.5 resulted in simulated ridges that matched the aspect ratio of ridges measured in Høyland, 2007. Is that a correct interpretation? Was the 1.5 value determined by iterating through different particle aspect ratios? Can you make any comments about the effect that particle aspect ratio has on the resultant ridge geometry? *- We reworded the sentence in question. We chose the aspect ratio so that it was possible for the ice to fail into blocks that could have the aspect ratios observed in the field. Discrete element size (defining the smallest fragment size) was, thus, chosen so that pieces could reach aspect ratios down to 1.5 observed in the field (Høyland, 2007).* 6. Lines 101-102 - Do you have any comments on how variable thickness may affect the simulated results, or how they might contribute to differences between the model and experiment? *- Hopkins et al. (1999) conducted two-dimensional simulations with ridges forming from variable ice thickness and state that the unevenness influences the ratio of ridging and rafting, but such study is out of the scope of the paper. We added this information to the manuscript as well.*

7. Equations 2 and 3 - Can you briefly explain why the scaling parameters have different exponents for the velocity term in Equation 2 and the force term in Equation 3? *- The scaling parameter adapts to the terms scaled. In Cauchy-Froude scaling all length scales (L) being scaled with $\lambda$, time (T) with $\lambda^{1/2}$ and mass (M) with $\lambda^3$. Thus, velocity results in $\mathrm{LT}^{-1} = \lambda^{1/2}$ and force in $\mathrm{MLT}^{-2} = \lambda^3$.*

8. Lines 138-143 - Both the simulated and experimental data seem like they are constantly increasing. Are there statistical tests that could show evidence of the two phases? Perhaps compute a moving window average and evaluate its slope in each phase? Or compute a regression line for each phase, and then compare the slopes of each? Along the same idea - you mention that the change from first to second phase is more pronounced in the simulations than in the experiments - do you have any comments or thoughts on why that is the case? *- We assume that the phases are less pronounced in the experimental data as not all the ice within the experiment always purely ridges, but rather rafts as well along the length of the ridge. Based on your suggestion, we looked into the distinguishing of the phases via applying running means to the data and then comparing the slopes of these running means. For the simulations, this analysis shows a clear change in slope and thus, we argue that the splitting in stages is justified. Our motivation is also partially included in the answer of point 9. For the experimental data, we decided to not split the data in stages and for the simulation data, we decided to include the analysis of the slope based on the running mean. Thus, Figure 3 changes in the manuscript and*

*additional information for the identification of the phases is included.*

9. It's not exactly clear to me why splitting the data into two phases is significant. Is the main idea that if the ridging force is more or less constant in the second phase, and that you can then use that to formulate some F h law? Assuming you do not need to split into two phases, could you use the maximum/peak ridging force instead of the mean force in the data fit? *- In essence, we repeated the analysis by Hopkins (1998): We split the force records the data into two periods similarly to Hopkins (1998) (and a conceptual model of ridging in Tuhkuri and Lensu (2002)). It is correct that then idea is that the second phase has quasi-constant force and the relationship between F and h is investigated based on this force. Similarly, Hopkins (1998) used the mean force for the fit. Following the suggestion, we also investigated the fit for five highest peak forces per simulation during $\delta = 40 \ldots 60$ m. The results did not change; linear fit describes the data. Information on this is added to the manuscript.*

10. Line 145 - How were the simulated ridge profiles computed? Referencing Figure 2b, the bottom surfaces look more "bumpy" than the profiles in Figure 4. Were the bottom-most particle positions sampled at some sort of regular interval along the width? *- Figure 4 shows mean ridge profiles, which is why they look less "bumpy". For the experiments, the dashed lines represent the individual profiles, while the solid line is the mean of the individual profiles. For the simulations, the solid line again is the mean of individual profiles, while the shading is the standard deviation. To compare the ridges with each other, all data was centered around the deepest point of the keel. Additionally, both datasets where "regridded" in regards to the width (x-axis). That means, that depth of the ridge is displayed along a width with a regular interval of 0.2m between each step. We treated both datasets the same for consistency. We highlighted the word mean more in regards to the ridge profiles from the simulations and added some explanation in the figure caption.*

11. Line 156 - W appears to continuously increase, and does not "plateau." *- We agree that it is not a constant plateau, but rather periods where ridges do not grow in width. We adapted the text accordingly.*

12. Line 196 - What is the "higher" in comparison to? 90% higher std dev than what simulation case? *- The comparision is between the simulations with $L = 10$ m to simulations with $L = 60$ m. We adapted the sentence to enhance clarity.*

13. Line 203 - Can you explain a little more what is meant by 'the setup as described above"? Do you mean your general simulation geometry? Or the general 3D DEM approach? Something else? *- ´the setup described above´ refers to the setup used for this study, which is described in section 2.2.1. We exchanged the word 'above' with a reference to the section.*

14. Lines 214-215 - Did you try to fit a FII h 3/2 type formula to this data? I would be curious to know what the Pearson coefficient is for that kind of fit. If you are arguing that a linear fit is more appropriate, then it makes sense to show the correlation coefficient of that fit, too, for comparison. The tail end of the data in Figure 7 appear to trend above the linear fit - did you run any simulations with thicker ice? It may be interesting to see if the F h relationship holds as h increases. *- We tested a fit with $h^{3/2}$ resulting in a Pearson coefficient of 0.99. We added this information to the manuscript as well as well as our justification to argue for the*

*linear fit, based on the lower polynomial order. We did not run any simulations with thicker ice. We already extended the thickness compared to the initial validation and think that with these thicknesses we cover the majority of typical level ice thicknesses occurring.*

15. Lines 218-219 - You reference your 2022 study that showed a linear relationship between ice load and thickness for simulations of ice against a rigid cone structure. Can you comment on the novelty of finding a similar relationship in this current manuscript? The larger floe in this paper was also modeled as a rigid structure, so the simulation setup seems fairly similar to the 2022 paper. *- The upwards-bending cone in the 2022 study had a diameter of 11 m. This is a narrow structure, which we included now into the text as well, and the ice failure process is different from here: In the case of a cone the process is inherently three-dimensional, whereas ridging has traditionally been considered a two-dimensional problem. Also in ridging, clearing of ice rubble does not occur.*

**Grammar/Spelling Corrections**

Suggested corrections are indicated in **bold**. 1. Line 1 - "...discrete element method simulations **of** pressure ridge formation." *- Changed.*

2. Line 21 - "**The** first theoretical models for..." *- Changed.*

3. Lines 76-77 - "We validated our simulations by comparing **the modeled** ridging force **magnitudes** and ridge profiles to those **measured in** the laboratory-scale experiments by Tuhkuri and Lensu (2002)." *- Changed.*

*ADDITIONAL REFERENCES*
*Ranta, J. and Polojärvi, A. (2019). Limit mechanisms for ice loads on inclined structures: Local crushing. Marine Structures, 67, 102633.*

---

## Author Response (AR1)

Dear Editor and Referees,

We sincerely thank everyone for their time, constructive feedback and insightful comments. As the answers to the reviewers have already been uploaded, we will compile here a list of last changes on the manuscript as uploaded. All changes in the manuscript are marked in blue in the track-changes file.

- The analysis of the phases is moved to the Appendix.

- Regarding the first reviewers question about the non-simultaneous failure (Line 191-197): In the reviewer answers we announced an additional Figure, but we decided against adding the additional Figure as Figure 2 already highlights non-simultaneous failure. We included a reference to accepted conference proceedings, where we compared the simulations with $L = 10$ m to simulations with $L = 60$ m as presented here. According to our knowledge, these are not published yet, but we are happy to forward the paper to the reviewers if requested.

To address the editor's suggestion to expand on the potential impact for future modelling studies, we added:

- L225: "related to the development of rheological models for pack ice and"

- L241: "; three dimensional DEM suggests that the ice strength, as often defined in continuum models, should increase linearly with $h$"

Additionally, the last paragraph of the conclusions (L261-267) expanded on this topic as well based on the reviewer comments.

Best regards,
Marek Muchow and Arttu Polojärvi

ADDITIONAL REFERENCES
Muchow, M. and Polojärvi, A.: Discrete Element Simulations on Pressure Ridge Formation: How the Length-Inclusion Facilitates New Research Avenues, Proceedings of the 27th IAHR International Symposium on Ice (electronic publication), accepted